# Presence of Trace Elements in Edible Insects Commercialized through Online E-Commerce Platform

**DOI:** 10.3390/toxics12100741

**Published:** 2024-10-12

**Authors:** Nadia San Onofre, David Vie, Jose M. Soriano, Carla Soler

**Affiliations:** 1Food & Health Lab, Institute of Materials Science, University of Valencia, 46980 Paterna, Spain; nsan_onofre@uoc.edu (N.S.O.); carla.soler@uv.es (C.S.); 2FoodLab Research Group, Faculty of Health Sciences, Universitat Oberta de Catalunya, Rambla del Poblenou 156, 08018 Barcelona, Spain; 3Institute of Materials Science, University of Valencia, 46980 Paterna, Spain; david.vie@uv.es; 4Joint Research Unit on Endocrinology, Nutrition and Clinical Dietetics, University of Valencia-Health Research Institute La Fe, 46026 Valencia, Spain

**Keywords:** edible insects, trace elements, food safety

## Abstract

This study aimed to evaluate the presence of various elements in edible insect-based food products available for human consumption. Several products were analyzed using atomic spectroscopy, and descriptive statistical analysis was conducted with IBM SPSS Statistics 27. The results revealed the presence of elements such as arsenic, cadmium, copper, magnesium, nickel, silver, lead, tungsten, uranium, mercury, platinum, aluminum, beryllium, bismuth, lithium, antimony, and thallium. Significant differences were found based on product type, insect species, and country of origin. The findings underscore the need to assess each insect species for its potential as a food source, taking into account element bioaccumulation factors. A comprehensive, global approach is essential for ensuring the food safety of edible insects as a sustainable protein source. Further research is needed to address these safety concerns.

## 1. Introduction

In recent years, insects have emerged as a nutritious, healthy, and sustainable alternative to meat, responding to the need to transform global food systems [1]. Insects are culturally accepted as food in several countries, such as Thailand, Mexico, China, and many African countries. However, much work remains to integrate them into European markets, where they are considered novel foods [2]. Due to their many benefits, insect consumption and incorporation into the food industry are experiencing global growth. There are clear environmental and economic advantages to replacing traditional animal protein sources with alternatives like insects, which require less feed, generate less waste, and produce fewer greenhouse gas emissions [3]. Additionally, the lower production costs of these new foods could impact their market price, potentially improving food security for vulnerable populations and promoting planetary health [4]. Despite progress in positioning insects in the international market, evidence reveals a series of barriers [5] that must be overcome to integrate them successfully. In the European Union, species such as *Acheta domestica* [6], *Alphitobius diaperinus* larvae [7], *Locusta migratoria* [8], and *Tenebrio molitor* [9] have been approved for human consumption and marketing. However, the European Food Safety Authority (EFSA) notes in its reports that production and consumption of insects as food must consider risks related to their allergenic potential and the possibility of containing harmful chemicals, such as environmental contaminants and trace elements. Research has shown that the toxicological profile of some insects depends on the environment in which they are raised and the food they receive [10]. However, much of the published research on edible insects focuses primarily on their nutritional value rather than on the contaminants they may contain. This research proposes the hypothesis that various metals are significantly present in edible insects and insect-based products. Transition metals in food play a crucial role in human nutrition: some are essential for biological processes, while others can be toxic. Essential transition metals, such as iron (Fe), zinc (Zn), and copper (Cu), are vital micronutrients involved in numerous physiological functions. Iron is necessary for oxygen transport and energy metabolism, zinc supports the immune system and aids in cell division and growth, and copper contributes to red blood cell formation and the maintenance of nerve cells and the immune system. However, excessive levels of these metals can cause toxicity. For example, excess iron can lead to organ damage, and high copper intake may result in liver and kidney damage. Conversely, some transition metals are toxic and unacceptable in food, even at low concentrations. These include cadmium, which can cause kidney damage and skeletal issues; lead, known for its neurotoxic effects, particularly in children; and mercury, which affects the nervous system and is especially harmful during pregnancy [11]. Detecting and quantifying these metals are crucial steps in assessing the global food safety of these products. This presents both challenges and opportunities in promoting insects as a sustainable protein source for human consumption. Therefore, the aim of this study is to assess the potential presence of trace metals in various food products made with insects and marketed for human consumption.

## 2. Materials and Methods

### 2.1. Chemical and Reagents

Internal standard solutions of Germanium (^72^Ge), Iridium (^193^Ir), Rhodium (^103^Rh), and Scandium (^45^Sc) at 20 µg/g, sourced from ISC Science (Oviedo, Spain), and certified standards were bought in the High-Purity Standards (North Charleston, SC, USA), being individual certified standards, including monitored isotopes in parentheses, for Arsenic (^75^As), Cadmium (^111^Cd), Copper (^63^Cu), Magnesium (^24^Mg), Nickel (^60^Ni), Silver (^107^Ag), Lead (^208^Pb), Tungsten (^182^W), Uranium (^238^U), Mercury (^201^Hg), and Platinum (^195^Pt) at 1000 mg/l, while individual certified standards for Aluminum (^27^Al), Beryllium (^9^Be), Bismuth (^209^Bi), Lithium (^7^Li), Antimony (^121^Sb), and Thallium (^205^Tl) at 1000 mg/L were incorporated from Scharlab (Scharlau, Barcelona, Spain). On the other hand, nitric acid, as 69% ultra-trace, ppb-trace analysis grade, was acquired from Scharlab, and hydrogen peroxide, for analysis, 35 wt% solution in deionized water, stabilized, arrived from Acros Organics (Geel, Belgium). Finally, deionized water (resistivity >18 MΩ cm^−1^) was obtained through a Milli-Q SP^®^ Reagent Water System (Millipore Corporation, Bedford, MA, USA).

### 2.2. Sampling and Sample Processing

All edible insects and insect-based foods (n = 31), commercially available and suitable for human consumption, were obtained from one E-Commerce Platform from December 2021 (Table 1). Upon receipt, the whole insects were immediately subjected to a drying process to eliminate water from the sample, in order to obtain results in ppm dry weight. This drying step was conducted at a controlled temperature of 60 °C in a drying oven (Binder FD 115, Tuttlingen, Germany) for 24 h, ensuring optimal conditions for subsequent milling. Once dried, the insects were ground into a fine powder using a laboratory mill (IKA M 20, Staufen, Germany). The mill was operated at a high speed, ensuring consistent particle size and homogeneity of the powder. Insect powder was used for each sample analysis (one aliquot per sample). This quantity was precisely weighed and prepared for mineralization and multielemental analysis. The powdered samples were stored in airtight containers at room temperature (+21 °C) to prevent any contamination or degradation prior to analysis.

### 2.3. Mineralization and Multielemental Analysis

For each analysis, 0.05 g of the powdered insect sample was accurately weighed and placed into Teflon mineralization vessels. The samples were then treated with 1 mL of hydrogen peroxide (H_2_O_2_, 30% *v*/*v*) and 4 mL of concentrated nitric acid (HNO_3_, 64% *v*/*v*) [12]. The mineralization was carried out using a microwave accelerated reaction system (MARS, CEM, Vertex, Spain) at 800 W and 180 °C for 15 min. This process ensured the complete mineralization of the sample, breaking down the organic matrix and releasing the metals into solution. After the mineralization process, the vessels were allowed to cool, and nitrogen vapors were safely eliminated. The digested samples were then filtered through Whatman No. 1 filter paper to remove any particulates, and made up to a final volume with deionized water. The resulting solutions were further diluted at a ratio of 1:5 with deionized water to prepare them for analysis.

An inductively coupled plasma spectrometer mass detector (ICP-MS, Agilent model 7900; Agilent Technologies, Tokyo, Japan)) was employed to identify and quantify the trace elements. The operating conditions were as follows: Ar plasma gas flow (15.0 L/min), carrier gas (1.07 L/min), reaction gas (He), nebulizer pump speed (0.10 rps), RF power (1550 W), and RF matching (1.80 V). Internal standard solutions of Ge, Rh, Ir, and Sc at 20 µg/g were used to correct matrix induced signal fluctuations and instrumental drift. Analysis was performed at Central Service for Experimental Research of the University of Valencia. Three measurements were taken for each sample. All of the equipment in the laboratory is certified with the norm ISO 9001:2015 [13], which guarantees a quality management system.

### 2.4. Statistical Analysis

A descriptive statistical analysis was performed on the analyzed samples, and the chemical results were compared using Student’s t tests, and a one-way analysis of variance (ANOVA) performed in SPSS Statistics for Windows was used for statistical analysis (version 27.0, Armonk, NY, USA, IBM Corp.). The level of statistical significance was set at *p* < 0.05 with a 95% confidence interval.

## 3. Results

Table 1 shows the studied samples. The study highlights a diverse array of samples originating from the United Arab Emirates, Thailand, the United Kingdom, and Europe, featuring species such as *Cyriopagopus albostriatus* (tarantulas), *Mesobuthus martensii* (scorpions), *Acheta domesticus* (crickets), *Gonimbrasia belina* (mopane worms), *Sphenarium purpurascens* (grasshoppers), *Bombyx mori* (silkworm pupae), *Atta cephalotes* (queen leafcutter ants), and *Tenebrio molitor* (mealworms).

Table 2 reveals significant variability in metal concentrations across various edible insect samples and insect-based products, indicating potential contamination concerns. Tarantula samples from the United Arab Emirates (CJ1) and Thailand (CJ24) show elevated levels of trace elements, while scorpion samples from the U.A.E. (CJ2, CJ3) and Thailand (CJ16, CJ25) also exhibit high levels, raising health concerns. Cricket-based products, such as flour from Thailand (CJ5) and the United Kingdom (CJ8, CJ12), and protein bars from Spain (CJ14, CJ27), display notable variations in metal content. Likewise, grasshopper samples from Mexico and the U.K. (CJ7, CJ10, CJ15, CJ17) suggest potential contamination risks. Although specific trace element profiles are not detailed, the variability implies regional environmental factors influencing metal accumulation. Mealworm samples from the Netherlands and the U.K. (CJ13, CJ18), along with silkworm pupae from Thailand (CJ9), show moderate metal concentrations. In terms of specific concentrations, magnesium levels range from 3229.42 ppm in silkworm pupae (CJ9) to 146.64 ppm in armor tail scorpions (CJ30), while aluminum peaks at 509.800 ppm in fried mopane worms (CJ6) and drops to less than 1 ppm in silkworm pupae (CJ9). Nickel reaches a maximum of 2.806 ppm in cricket chocolate bars (CJ28), with lower values like 0.089 ppm observed in silkworm pupae (CJ9). Arsenic levels (1.769 ppm in scorpions, CJ3) and lead concentrations (0.754 ppm in salt-flavored zebra tarantula, CJ24) are notable, with particularly high cadmium in the same tarantula sample (CJ24) at 2.643 ppm. Additionally, significant concentrations of silver and bismuth were detected, with 0.364 ppm silver and 0.214 ppm bismuth in the salt-flavored zebra tarantula (CJ24). Other metals, such as W, Tl, U, Be, Sb, Pt, and Hg, exhibit varying levels, with some samples, including the zebra tarantula (CJ24) and armor tail scorpions (CJ25), showing elevated concentrations, further suggesting potential contamination issues.

Table 3 provides an extensive statistical analysis of trace element concentrations in various edible insect products, revealing significant insights into their contamination levels.

The data presented in Table 3 show significant variability in metal concentrations across the analyzed samples, which likely reflects a combination of environmental contamination, industrial processing, and sourcing practices. For example, the wide range in lithium (Li), magnesium (Mg), and aluminum (Al) concentrations points to differences in production environments, and possibly the inclusion of these metals for manufacturing purposes. High levels of toxic elements such as cadmium (Cd), lead (Pb), mercury (Hg), and arsenic (As) raise serious public health concerns. Their presence indicates potential contamination from industrial pollution, or inadequate quality control during production. Given their well-known toxicological profiles, these metals, even at low concentrations, could pose long-term health risks to consumers, especially in more vulnerable populations. Furthermore, the variability in elements such as tungsten (W), silver (Ag), and bismuth (Bi), which appear in lower but inconsistent quantities, suggests differences in raw material sources or contamination during the manufacturing process. The presence of trace metals such as thallium (TI), uranium (U), and antimony (Sb), although in smaller quantities, further supports the likelihood of industrial contamination, and their detection necessitates continuous monitoring and regulatory intervention. Table 4 presents a detailed comparative analysis of trace element concentrations in edible insect products from different regions, America (n = 5), Asia (n = 11), and Europe (n = 15), highlighting significant regional variations.

Lithium (Li) levels are notably highest in products from Asia, with a mean of 0.108 ppm, significantly differing from Europe (0.039 ppm) and America (0.107 ppm), with significant *p*-values (*p* = 0.005 for Asia, *p* < 0.001 for Europe and America, ANOVA *p* = 0.026). Magnesium (Mg) content is also elevated in Asian samples (1052.680 ppm) compared to European (979.110 ppm) and American products (516.360 ppm), though the ANOVA P-value is less significant (*p* = 0.391). Aluminum (Al) shows a marked difference, with Asian samples having a mean of 144.300 ppm, higher than European (56.870 ppm) and American products (100.580 ppm), with significant *p*-values (*p* = 0.006, ANOVA *p* = 0.218). Nickel (Ni) levels show variability, with European products having the highest mean (0.578 ppm) and American samples the lowest (0.483 ppm, *p* = 0.907). Tungsten (W) and arsenic (As) levels are also significantly higher in Asian products (0.507 ppm for W and 0.541 ppm for As), with *p*-values indicating regional differences (*p* = 0.003 for W, *p* = 0.014 for As, ANOVA *p* = 0.204 and *p* = 0.012, respectively). Silver (Ag), cadmium (Cd), and lead (Pb) are also found in higher concentrations in Asian products, with mean values of 0.238 ppm for Ag, 1.475 ppm for Cd, and 0.209 ppm for Pb, compared to significantly lower levels in European and American samples (*p* < 0.005, ANOVA *p* < 0.001). Thallium (TI), bismuth (Bi), uranium (U), and antimony (Sb) also exhibit regional variations, with Asian products having generally higher levels, likely reflecting differences in environmental contamination and processing. Mercury (Hg) concentrations further highlight regional disparities, with significantly higher levels in Asian samples (0.252 ppm) compared to European (0.054 ppm) and American samples (0.061 ppm, *p* = 0.002, ANOVA *p* = 0.002). The data suggest that edible insect products from Asia exhibit higher contamination levels for several metals, pointing to the need for region-specific monitoring and regulation to address potential health risks associated with trace element exposure in these products.

## 4. Discussion

The increasing incorporation of insects into the human diet has sparked significant interest in their food safety and nutritional quality. The Food and Agriculture Organization of the United Nations [14] has highlighted the importance of assessing the risks associated with insect consumption, focusing on three main areas: allergenicity, microbiological risks, and chemical risks. In light of these concerns, this research serves as a response to this emerging trend. The primary goal of this study was to conduct a chemical analysis of edible insects to identify the potential presence of metals and environmental contaminants. Our results revealed a notable diversity in the concentrations of various chemical elements. Poma et al. [15] demonstrated that farmed insects have the potential to accumulate chemicals, and Malematja et al. [16] further explored this phenomenon, linking metal presence in insects to their diet, which was based on agricultural waste. This finding underscores the role of diet in trace element accumulation in insects, with potential implications for food safety. Despite these findings, the health risks posed by these chemical elements remain unclear. While some studies report low contaminant levels and conclude that insect consumption does not represent an additional risk compared to animal-based products [15], others have found trace element [17] and biological contaminant levels [18] that could potentially be harmful to human health. Given the varying results in the existing research, it is essential to continue exploring this field to safeguard food security and fully understand the risks associated with this novel food source, which is gaining popularity worldwide. Factors to consider include the insects’ food sources and the presence of chemical contaminants in insect-based products. Moreover, there is an urgent need to translate theoretical evidence into practice through more precise regulations for monitoring the production and consumption of insect-based foods. Such regulations should provide assurances to producers while maximizing consumer protection [19]. Within this study, we observed significant variability in metal concentrations across analyzed products.

When we examined results by insect group, we found substantial differences in elemental concentrations, except for *Mesobuthus martensii*. Elements like Li, Al, As, W, Pb, Bi, U, and Hg showed no significant differences in products containing *M. martensii*, prompting reflections on both the biology of this species and the implications for food safety regulations. This could suggest that *M. martensii* has mechanisms for metal accumulation, tolerance, or self-regulation, as seen in other insect species [20]. These mechanisms could relate to their natural habitat or diet, indicating the need for further research into their biology and environmental interactions. Other researchers have described patterns of metal and contaminant accumulation, noting that bioaccumulation patterns vary by insect species and developmental stage [21]. The consistency in elemental presence in *M. martensii* has implications for food safety and public health. If products containing *M. martensii* exhibit more consistent elemental concentrations, they could offer more reliable quality and safety compared to products made from other insect species. This consistency could inform regulations and labeling for insect-based food products.

Research on *Acheta domesticus* also shows variability, revealing various toxic compounds in these products [22]. Studies found concentrations of elements such as As (0.01–0.08 ppm), Al (34 ppm), Be (0.01–0.02 ppm), Cd (0.01–0.02 ppm), Hg (0.038–0.041 ppm), Li (0.01–0.04 ppm), Ni (0.13–0.32 ppm), Pb (0.06–0.2 ppm), Sb (n.d.-0.83 ppm), and Ti (0.09–0.14 ppm) with considerable variability depending on the sample analyzed, highlighting the need for stricter controls to ensure product quality and food safety [21]. Pastell et al. [23] detected chromium and nickel in various samples of *A. domesticus*. In our study, we identified 17 metals in *A. domesticus* and its food products, with notable concentrations of Mg, Ni, and W. Although the EFSA [7] considers *A. domesticus* safe for consumption, further research is needed to understand these differences within the broader scientific context.

As previously mentioned, insects are an integral part of the gastronomic culture in many regions. In countries like Uganda [24], where grasshoppers and other insects are commonly consumed, their role as a protein source has been promoted, even though they have been identified as vectors for metal exposure, particularly Pb. Studies by Butt et al. [25] and Kasozi et al. [26] found that grasshoppers accumulated more Pb than Cd and Zn, with varying concentrations observed in other insect species. In contrast, Bednarska et al. [27] studied crickets—closely related to grasshoppers and locusts—and found that crickets regulate zinc intake more efficiently than cadmium, indicating a tendency to accumulate cadmium.

Janssen et al. [28] indicated that seasonal variations in metal concentrations, along with developmental stage differences, could play a fundamental role in bioaccumulation and must be considered when analyzing concentration differences. Similarly, the available information on other trace elements, such as aluminum, chromium, and arsenic, remains scarce, highlighting the need for further research [29]. In entomology, studies have demonstrated the capacity of metals to transfer along food chains involving insects. Such studies have conclusively shown that the feeding behavior of these insects substantially influences metal bioaccumulation [25]. Si et al. [30] examined metal accumulation and movement through a food chain involving soil, mulberry trees, and *Bombyx mori*, in response to exposure to lead and cadmium, both individually and in combination. In their study, mulberry leaves contaminated with trace elements were fed to silkworms, which led to a notable increase in trace element concentration within the silkworms. However, a significant portion of these metals was excreted, resulting in a concentration of trace elements within the silkworms that was lower than the amount ingested. In the present study, we observed that the concentrations of Li, Al, and U were below detection limits, while the highest levels were of Mg, As, and W. Notably, no significant concentrations of Pb or Cd were detected. Studies on ants [31] have found bioaccumulation of Cd, K, P, S, and Zn in quantities significantly greater than in other organisms, although the element concentrations in ants remained stable regardless of soil concentrations. This suggests that ants possess regulatory mechanisms for maintaining stable element concentrations, potentially linked to their biological adaptations and ecological roles. However, further research is needed to fully understand these mechanisms and their implications for food safety. In our research, the elements found in the highest concentrations were Mg, W, Al, and Li. This finding deviates from previous research [31], suggesting a need for further study to better understand bioaccumulation in these insect species. Regarding *Tenebrio molitor*, research by Lindqvist and Block [32] and Bednarska et al. [27] has shown a positive correlation between Cd concentrations in T. molitor and the levels of Cd present in its diet. However, while *T. molitor* may absorb Cd, this does not necessarily indicate harm, as metals can be excreted metabolically, reducing their potential toxicity by limiting their interaction with receptor sites in the body. Consequently, only a fraction of the total Cd load in the body is available to exert toxic effects [33]. Truzzi et al. [12] showed that metal concentrations detected in T. molitor were below legal limits, indicating that the risk of metal exposure from consuming mealworm larvae is relatively low. In our investigation, we detected 17 elements, with higher concentrations of Mg, Ni, As, and W, which aligns with an EFSA report stating that trace element concentrations in T. molitor are comparable to those in other foods. Furthermore, current EU regulations do not specify maximum limits for trace elements in insects for human consumption. Based on these findings, the consumption of *T. molitor* within proposed parameters is considered safe, although it may pose a risk of allergic reactions in certain individuals [10]. This study also included analysis of *Alphitobius diaperinus*, which is approved as a safe novel food in Europe [8], although no prior studies have examined its metal bioaccumulation potential. In our study, *A. diaperinus* showed the highest concentrations of Mg, followed by Ni and W. We also examined *Haplopelma albostriatum* and *Gonimbrasia belina*, which are not yet approved by the EFSA for human consumption. *G. belina*, widely consumed in parts of Africa, is primarily studied for its nutritional value rather than food safety [34]. In this investigation, *G. belina* displayed the highest concentrations of Mg, Cd, and Ag. Regional differences in element concentrations were also observed; specifically, products from Asia exhibited significantly higher levels of Li, Al, Cu, As, Ag, Cd, Pb, U, and Hg. This raises questions regarding the sources of these metals and their potential influence on product quality. Conversely, no statistically significant differences were found in other element concentrations, although Ni was most concentrated in European products, while Be, Sb, and Pt were highest in products from the UK. These patterns may reflect geological or industrial factors specific to these regions, warranting further investigation. Overall, the highest concentrations of elements were observed in products from Asia, prompting concerns about contamination sources and production practices in this region. These findings have implications not only for product quality, but also for public health and food regulation.

The statistically significant variability (*p* < 0.001) across all metal concentrations emphasizes the critical need for comprehensive assessment protocols to manage and mitigate the risks associated with trace element exposure in the growing edible insect industry. In light of the current study on trace elements in edible insects, it is important to discuss the potential implications of toxic metal contamination in food products. Previous research has shown that maintaining the stability of raw materials through methods like water activity control is essential to preventing contamination and preserving product quality [35]. Additionally, the impact of metallic nanoparticles, such as titanium dioxide, on gut fermentation has highlighted potential health risks associated with nanoparticle ingestion, underscoring the importance of monitoring toxic elements in food [36]. Moreover, the role of toxic metals like cadmium and mercury in metabolic disorders, including obesity, has been well documented, reinforcing the need for careful consideration of metal exposure through food sources [37]. Finally, studies on heavy metal contamination in dried fish from Bangladesh emphasize the health risks posed by high levels of toxic elements, which are relevant for the discussion of insect-based food products and the need for stringent safety standards [38]. These findings collectively highlight the importance of addressing and mitigating potential toxic metal exposure in edible insect products to ensure consumer safety.

On the other hand, presence of trace elements from insect-based food products have been reflected in the literature; i.e., Sikora et al. [39] detected contents of Al and Pb in feed based on yellow mealworm, and Gori et al. [40] observed concentrations of Pb, Cd, and Ni in insect-based products sold by e-commerce in the EU market.

Further studies are needed to better understand the underlying causes of these regional differences in metal concentrations and their impact on food safety and consumer health. The FAO emphasizes the importance of considering factors such as metal type, insect species, growth stage, substrates used, and environmental contamination, as all of these have been linked to metal accumulation in edible insects [13,41]. Although this study analyzed 31 insect-based products—which do not represent the full range of products available on the international market—it highlights the need for comprehensive research on the nutritional suitability of all edible insects to ensure global food security that serves the entire population, irrespective of geographical distinctions. A lack of scientific data related to various insects consumed in regions of Africa, Asia, and the Americas underscores the urgency of expanding the knowledge base in this field. Ensuring a broad and equitable approach is essential to understanding the spectrum of insects consumed in different parts of the world.

## 5. Conclusions

In conclusion, edible insects are a novel food source in Europe [42] and offer a sustainable alternative to meat-based proteins due to their high nutritional value, providing essential macro- and micronutrients. The chemical analysis conducted in this study confirmed the presence of metals and trace elements in all insect samples analyzed, raising food safety concerns—particularly for vulnerable groups such as children, pregnant women, and individuals with health conditions, which must be carefully considered. This study underscores the need to evaluate each insect species specifically for its suitability as a food source, accounting for factors related to metal bioaccumulation. This bioaccumulation varies by insect species and geographic origin, emphasizing the importance of a global perspective in assessing food safety.

Additionally, differences in metal concentrations were observed between insects from various regions, highlighting the importance of considering the origin of insects when evaluating their safety as food. This study also points out the lack of data on the bioaccumulation of certain chemical elements in specific insect species, particularly in regions of Africa, Asia, and parts of the Americas, underlining the need for further research to gain a comprehensive understanding of this issue. In light of the results, it is crucial to establish stronger regulations for insect consumption as a food source, and to define safe limits for the presence of trace elements in edible insects. Such regulations will help protect consumer health as insects become an increasingly popular protein source.

## Figures and Tables

**Table 1 toxics-12-00741-t001:** Characteristics of studied edible insects (*n* = 13) and insect-based foods (n = 18).

Code	Sample	Origin	Analyzed Species	% Insect
CJ1	Tarantula ^1^	United Arab Emirates (Asia)	*Cyriopagopus albostriatus*	100
CJ2	Scorpion ^1^	United Arab Emirates (Asia)	*Mesobuthus martensii*	100
CJ3	Scorpion ^1^	United Arab Emirates (Asia)	*Mesobuthus martensii*	100
CJ4	Nutritional pasta ^2^	Spain (Europe)	*Acheta domesticus*	Nr
CJ5	Cricket flour ^1^	Thailand (Asia)	*Acheta domesticus*	100
CJ6	Mopane worms (fried) ^1^	United Kingdom (Europe)	*Gonimbrasia belina*	97
CJ7	Lemon grasshoppers ^2^	Mexico (America)	*Sphenarium purpurascens*	87
CJ8	Cheese and onion crickets ^2^	United Kingdom (Europe)	*Acheta domesticus*	87
CJ9	Silkworm pupae ^1^	Thailand (Asia)	*Bombyx mori*	100
CJ10	Lemon grasshoppers ^2^	Mexico (America)	*Sphenarium purpurascens*	87
CJ11	Queen leafcutter ants ^1^	Colombia (America)	*Atta cephalotes*	99.5
CJ12	Salt and vinegar crickets ^1^	United Kingdom (Europe)	*Acheta domesticus*	87
CJ13	Kids critters mealworms and crickets ^2^	Netherlands and United Kingdom (Europe)	*Tenebrio molitor* + *Acheta domesticus*	70 + 30
CJ14	Cricket protein bar ^2^	Spain (Europe)	*Acheta domesticus*	5
CJ15	Garlic grasshoppers ^2^	United Kingdom (Europe)	*Sphenarium purpurascens*	83
CJ16	Armor tail scorpions ^1^	Thailand (Asia)	*Mesobuthus martensii*	86
CJ17	Chilli grasshoppers ^2^	Mexico (America)	*Sphenarium purpurascens*	73
CJ18	BBQ mealworms ^2^	Netherland and United Kingdom (Europe)	*Tenebrio molitor*	87
CJ19	Chilli grasshoppers ^2^	Mexico (America)	*Sphenarium purpurascens*	73
CJ20	Flour ^1^	Belgium (Europe)	*Acheta domesticus*	97
CJ21	Jungle trail mix ^1^	Thailand (Asia)	*Heterometrus longimanus*, *Lethocerus indicus and Mesobuthus martensii*	100
CJ22	Chocolate ball with cricket ^2^	Finland (Europe)	*Acheta domesticus*	7.5
CJ23	Garlic grasshoppers ^2^	United Kingdom (Europe)	*Sphenarium purpurascens*	83
CJ24	Salt flavor zebra tarantula ^2^	Thailand (Asia)	*Cyriopagopus albostriatus*	<100
CJ25	Armor tail scorpions ^1^	Thailand (Asia)	*Mesobuthus martensii*	Nr
CJ26	Chocolate covered grasshoppers ^2^	Thailand (Asia)	*Sphenarium purpurascens*	Nr
CJ27	Cricket protein bar ^2^	Spain (Europe)	*Acheta domesticus*	5
CJ28	Cricket chocolate bar ^2^	Spain (Europe)	*Acheta domesticus*	5
CJ29	Pasta ^2^	Germany (Europe)	*Alphitobius diaperinus*	14
CJ30	Armor tail scorpions ^1^	Thailand (Asia)	*Mesobuthus martensii*	Nr
CJ31	Chocolate ball with cricket ^2^	Finland (Europe)	*Acheta domesticus*	7.5

Nr = Not reflected. ^1^ Edible insect. ^2^ Insect-based food.

**Table 2 toxics-12-00741-t002:** Measured levels of trace elements in studied edible insects and insect-based food.

Sample	Li (ppm)	Mg (ppm)	Al (ppm)	Ni (ppm)	Cu (ppm)	As (ppm)	Ag (ppm)	Cd (ppm)	W (ppm)	TI (ppm)	Pb (ppm)	Bi (ppm)	U (ppm)	Be (ppm)	Sb (ppm)	Pt (ppm)	Hg (ppm)
CJ1	0.21 ± 0.01 f	1337.6 ± 0.3 c	268 ± 3 d	0.48 ± 0.01 b	65 ± 1 c	0.32 ± 0.01 b	0.14 ± 0.01 b	1.61 ± 0.01 c	1.43 ± 0.01 f	0.03 ± 0.01 d	0.500 ± 0.002 c	0.21 ± 0.01 f	0.03 ± 0.01 d	0.02 ± 0.01 d	0.03 ± 0.01 d	0.02 ± 0.1 e	0.54 ± 0.01 e
CJ2	0.15 ± 0.01 d	515.8 ± 0.7 b	276 ± 2 d	0.53 ± 0.01 b	103.9 ± 0.70 d	0.50 ± 0.01 c	0.22 ± 0.01 b	1.90 ± 0.01 d	0.92 ± 0.01 e	0.02 ± 0.01 c	0.176 ± 0.004 b	0.13 ± 0.01 d	0.04 ± 0.01 e	<0.01 a	0.01 ± 0.01 b	0.01 ± 0.01 d	0.66 ± 0.01 f
CJ3	0.34 ± 0.01 g	757 ± 1 b	204 ± 2 c	0.55 ± 0.01 b	216 ± 2 f	1.77 ± 0.01 d	0.75 ± 0.01 e	4.53 ± 0.01 f	1.03 ± 0.01 e	0.06 ± 0.01 e	0.488 ± 0.002 c	0.17 ± 0.01 e	0.04 ± 0.01 e	0.01 ± 0.01 b	0.01 ± 0.01 b	0.01 ± 0.01 c	0.31 ± 0.01 c
CJ4	0.04 ± 0.01 b	574 ± 2 b	11 ± 5 a	0.11 ± 0.01 a	4 ± 3 a	0.09 ± 0.01 a	0.02 ± 0.01 a	0.05 ± 0.01 a	0.62 ± 0.01 d	0.01 ± 0.01 a	0.038 ± 0.006 a	0.09 ± 0.01 c	0.01 ± 0.01 a	<0.01 a	<0.01 a	0.01 ± 0.01 c	0.12 ± 0.01 b
CJ5	0.05 ± 0.01 b	1002 ± 1 c	28 ± 3 a	0.47 ± 0.01 b	25 ± 2 a	0.15 ± 0.01 a	0.02 ± 0.01 a	0.04 ± 0.01 a	0.65 ± 0.01 d	0.01 ± 0.01 a	0.189 ± 0.003 b	0.09 ± 0.01 c	0.01 ± 0.01 a	0.01 ± 0.01 a	0.03 ± 0.01 e	0.01 ± 0.01 d	0.13 ± 0.01 b
CJ6	0.06 ± 0.01 b	1903 ± 2 d	510 ± 2 f	0.45 ± 0.01 b	4 ± 2 a	0.14 ± 0.01 a	0.01 ± 0.01 a	0.06 ± 0.01 a	0.61 ± 0.01 d	0.02 ± 0.01 c	0.205 ± 0.005 b	0.08 ± 0.01 c	0.03 ± 0.01 d	<0.01 a	<0.01 a	0.01 ± 0.01 c	0.10 ± 0.01 b
CJ7	0.05 ± 0.01 b	300.9 ± 0.7 a	78 ± 1 b	0.38 ± 0.01 b	13 ± 1 a	0.06 ± 0.01 a	0.01 ± 0.01 a	0.10 ± 0.01 a	0.56 ± 0.01 c	0.01 ± 0.01 a	0.135 ± 0.003 b	0.08 ± 0.01 c	0.01 ± 0.01 b	0.011 ± 0.001 b	0.03 ± 0.01 d	0.01 ± 0.01 d	0.08 ± 0.01 a
CJ8	0.03 ± 0.01 a	607 ± 1 b	16 ± 4 a	0.29 ± 0.01a	24 ± 1 a	0.16 ± 0.01 a	<0.01 a	<0.03 a	0.49 ± 0.01 c	0.01 ± 0.01 a	0.041 ± 0.006 a	0.07 ± 0.01 c	0.02 ± 0.01 c	0.01 ± 0.01 b	0.03 ± 0.01 d	0.01 ± 0.01 c	0.09 ± 0.01 a
CJ9	<0.02 a	3229 ± 1 f	<1.00 a	0.09 ± 0.01 a	10 ± 2 a	0.53 ± 0.01 c	0.02 ± 0.01 a	0.03 ± 0.01 a	0.41 ± 0.01 b	0.01 ± 0.01 a	0.084 ± 0.005 a	0.05 ± 0.01 b	0.01 ± 0.01 b	<0.01 a	0.02 ± 0.01 c	0.01 ± 0.01 c	0.06 ± 0.01 a
CJ10	0.05 ± 0.01 b	508 ± 2 a	103 ± 2 b	0.61 ± 0.01 b	15 ± 1 a	<0.05 a	0.01 ± 0.01 a	0.17 ± 0.01 a	0.46 ± 0.01 c	0.01 ± 0.01 a	0.147 ± 0.003 b	0.05 ± 0.01 b	0.01 ± 0.01 b	<0.01 a	0.02 ± 0.01 c	0.01 ± 0.01 d	0.07 ± 0.01 a
CJ11	0.13 ± 0.01 d	651.8 ± 0.8 b	150 ± 3 c	0.09 ± 0.01 a	12 ± 2 a	<0.05 a	0.03 ± 0.01 a	<0.03 a	0.38 ± 0.01 b	0.01 ± 0.01 a	0.061 ± 0.005 a	0.04 ± 0.01 b	0.01 ± 0.01 b	0.011 ± 0.001b	0.02 ± 0.01 d	0.01 ± 0.01 d	0.08 ± 0.01 a
CJ12	<0.02 a	655 ± 1 b	17 ± 3 a	0.26 ± 0.01 a	21 ± 2 a	<0.05 a	0.01 ± 0.01 a	0.04 ± 0.01 a	0.36 ± 0.01 b	0.01 ± 0.01 a	0.12 ± 0.01 b	0.04 ± 0.01 b	0.01 ± 0.01 b	0.014 ± 0.001 b	0.03 ± 0.01 d	0.01 ± 0.01 d	0.07 ± 0.01 a
CJ13	0.01 ± 0.01 a	1735.2 ± 0.8 d	9 ± 8 a	0.48 ± 0.01 b	19 ± 2 a	0.22 ± 0.01 b	0.01 ± 0.01 a	0.08 ± 0.01 a	0.31 ± 0.01 b	0.01 ± 0.01 a	0.02 ± 0.01 a	0.03 ± 0.01 a	0.01 ± 0.01 b	<0.01 a	0.01 ± 0.01 b	0.01 ± 0.01 c	0.05 ± 0.01 a
CJ14	0.04 ± 0.01 b	581 ± 1 b	7 ± 3 a	0.50 ± 0.01 b	3 ± 1 a	0.10 ± 0.015 a	0.01 ± 0.01 a	0.03 ± 0.01 a	0.29 ± 0.01 b	0.005 ± 0.008 a	0.03 ± 0.01 a	0.04 ± 0.01 b	0.01 ± 0.01 b	<0.01 a	0.01 ± 0.01 b	0.01 ± 0.01 c	0.04 ± 0.01 a
CJ15	0.12 ± 0.01 d	559.5 ± 0.7 b	105 ± 3 b	0.57 ± 0.01 b	15.6 ± 0.9 a	0.14 ± 0.01 a	0.02 ± 0.01 a	0.13 ± 0.01 a	0.31 ± 0.01 b	0.02 ± 0.01 c	0.14 ± 0.01 b	0.04 ± 0.01 b	0.01 ± 0.01 b	0.05 ± 0.01 e	0.06 ± 0.01 f	0.03 ± 0.01 f	0.14 ± 0.01 b
CJ16	0.03 ± 0.01 a	226.6 ± 0.8 a	20 ± 4 a	0.22 ± 0.01 a	152 ± 1 e	0.27 ± 0.01 b	0.464 ± 0.003 d	1.60 ± 0.01 c	0.25 ± 0.01 a	0.007 ± 0.007 a	0.07 ± 0.01 a	0.05 ± 0.01 b	0.01 ± 0.01 b	0.01 ± 0.01 b	0.01 ± 0.01 b	0.01 ± 0.01 c	0.15 ± 0.01 b
CJ17	0.15 ± 0.01 d	564.4 ± 0.9 b	84 ± 3 b	0.62 ± 0.01 b	16 ± 2 a	0.10 ± 0.01 a	0.01 ± 0.01 a	0.13 ± 0.01 a	0.31 ± 0.01 b	0.02 ± 0.01 c	0.16 ± 0.01 b	0.05 ± 0.01 b	0.01 ± 0.01 b	0.010 ± 0.001 b	0.02 ± 0.01 c	0.01 ± 0.01 b	0.04 ± 0.01 a
CJ18	0.03 ± 0.01 a	1976 ± 1 d	13 ± 6 a	0.46 ± 0.01 b	10 ± 3 a	0.27 ± 0.01 b	0.02 ± 0.01 a	0.12 ± 0.01 a	0.27 ± 0.01 b	0.013 ± 0.006 b	0.02 ± 0.01 a	0.06 ± 0.01 b	0.01 ± 0.01 b	0.01 ± 0.01 b	0.02 ± 0.01 c	0.01 ± 0.01 c	0.03 ± 0.01 a
CJ19	0.16 ± 0.01 e	557.05 ± 1.00 b	88 ± 2 b	0.71 ± 0.01 c	16 ± 1 a	0.13 ± 0.01 a	0.02 ± 0.01 a	0.16 ± 0.01 a	0.24 ± 0.01 a	0.02 ± 0.01 c	0.15 ± 0.01 b	0.03 ± 0.01 a	0.01 ± 0.01 b	0.015 ± 0.004 c	0.02 ± 0.01 d	0.01 ± 0.01 b	0.04 ± 0.01 a
CJ20	0.03 ± 0.01 a	1246 ± 1 c	8 ± 3 a	0.26 ± 0.01 a	23 ± 1 a	0.10 ± 0.01 a	0.01 ± 0.01 a	0.13 ± 0.01 a	0.16 ± 0.01 a	0.007 ± 0.009 a	0.04 ± 0.01 a	0.02 ± 0.01 a	0.01 ± 0.01 b	<0.01 a	0.01 ± 0.01 b	0.01 ± 0.01 c	0.03 ± 0.01 a
CJ21	0.11 ± 0.01 c	570 ± 4 b	385 ± 1 e	0.43 ± 0.01 b	8 ± 1 a	1.69 ± 0.01 d	0.04 ± 0.01 a	1.33 ± 0.01 c	0.22 ± 0.01 a	0.01 ± 0.01 a	0.69 ± 0.01 d	0.03 ± 0.016 a	0.04 ± 0.01 e	0.015 ± 0.010 c	0.02 ± 0.01 c	0.01 ± 0.01 b	0.41 ± 0.01 d
CJ22	<0.02 a	615.0 ± 0.7 b	9 ± 7 a	0.67 ± 0.01 c	5 ± 1 a	0.06 ± 0.01 a	<0.01 a	<0.03 a	0.23 ± 0.01 a	0.01 ± 0.01 a	0.01 ± 0.01 a	0.02 ± 0.01 a	0.001 ± 0.001 a	<0.01 a	0.01 ± 0.01 b	0.01 ± 0.01 b	0.03 ± 0.01 a
CJ23	0.08 ± 0.01 c	558.6 ± 0.8 b	102 ± 3 b	0.61 ± 0.01 b	2 ± 2 a	<0.05 a	0.01 ± 0.01 a	0.10 ± 0.01 a	0.18 ± 0.01 a	0.01 ± 0.01 b	0.13 ± 0.01 b	0.03 ± 0.01 a	0.003 ± 0.002 a	<0.01 a	<0.01 a	0.01 ± 0.01 b	0.02 ± 0.01 a
CJ24	0.14 ± 0.01 d	2588 ± 1 e	250 ± 2 d	0.33 ± 0.01 a	150 ± 2 e	0.20 ± 0.01 a	0.36 ± 0.01 c	2.64 ± 0.01 e	0.13 ± 0.01 a	0.01 ± 0.01 b	0.75 ± 0.01 d	0.02 ± 0.01 a	0.05 ± 0.01 f	<0.01 a	0.01 ± 0.01 b	<0.005 a	0.33 ± 0.01 c
CJ25	0.10 ± 0.01 c	350.4 ± 0.6 a	92 ± 3 b	0.32 ± 0.01 a	106 ± 2 d	0.34 ± 0.01 b	0.38 ± 0.01 c	1.84 ± 0.01 d	0.16 ± 0.01 a	0.008 ± 0.005 a	0.13 ± 0.01 b	0.02 ± 0.01 a	0.01 ± 0.01 b	0.01 ± 0.01 b	0.01 ± 0.01 b	0.01 ± 0.01 b	0.09 ± 0.005 a
CJ26	0.02 ± 0.01 a	855.8 ± 0.7 b	31 ± 4 a	1.87 ± 0.01 d	7 ± 2 a	0.08 ± 0.01 a	0.016 ± 0.012 a	0.10 ± 0.01 a	0.23 ± 0.01 a	0.01 ± 0.01 a	0.03 ± 0.01 a	0.02 ± 0.01 a	0.003 ± 0.001 a	0.01 ± 0.01 b	0.02 ± 0.01 c	0.01 ± 0.01 b	0.03 ± 0.01 a
CJ27	0.04 ± 0.01 b	393 ± 1 a	6 ± 4 a	0.25 ± 0.01 a	1.9 ± 0.9 a	0.08 ± 0.01 a	0.02 ± 0.01 a	0.03 ± 0.01 a	0.17 ± 0.03 a	<0.005 a	0.02 ± 0.01 a	0.02 ± 0.01 a	0.007 ± 0.003 a	0.01 ± 0.01 b	0.01 ± 0.01 b	0.01 ± 0.01 b	0.02 ± 0.01 a
CJ28	0.01 ± 0.01 a	2065.1 ± 0.9 d	17 ± 6 a	2.81 ± 0.01 e	13.6 ± 0.9 a	0.05 ± 0.01 a	<0.01 a	0.6 ± 0.1 b	0.13 ± 0.01 a	0.01 ± 0.01 a	0.02 ± 0.01 a	0.01 ± 0.01 a	0.002 ± 0.001 a	<0.01 b	0.01 ± 0.01 b	0.01 ± 0.01 c	0.02 ± 0.01 a
CJ29	0.02 ± 0.01 a	577 ± 1 b	11 ± 8 a	0.18 ± 0.01 a	5 ± 1 a	0.08 ± 0.01 a	<0.01 a	0.02 ± 0.01 a	0.11 ± 0.01 a	<0.005 a	0.02 ± 0.01 a	0.01 ± 0.01 a	0.002 ± 0.001 a	<0.01 b	<0.01 b	0.01 ± 0.01 b	0.02 ± 0.01 a
CJ30	0.02 ± 0.01 a	147 ± 2 a	28 ± 4 a	0.16 ± 0.01 a	34 ± 2 b	0.10 ± 0.01 a	0.20 ± 0.01 b	0.6 ± 0.01 b	0.15 ± 0.01 a	<0.005 a	0.08 ± 0.01 a	0.03 ± 0.01 a	0.006 ± 0.004 a	<0.01 b	0.01 ± 0.01 b	0.01 ± 0.01 b	0.07 ± 0.01 a
CJ31	0.03 ± 0.01 a	641 ± 1 b	12 ± 5 a	0.75 ± 0.01 c	3 ± 2 a	<0.05 a	0.03 ± 0.01 a	<0.03 a	0.17 ± 0.01 a	0.01 ± 0.01 a	0.02 ± 0.01 a	0.02 ± 0.01 a	<0.001 a	0.01 ± 0.01 b	<0.01 a	0.01 ± 0.01 b	0.02 ± 0.01 a

Different letters in columns indicate statistically significant differences among samples for the same element.

**Table 3 toxics-12-00741-t003:** Descriptive statistics of the studied compounds in the analyzed products.

Element (ppm)	Minimum–Maximum	Mean ± SD	Standard Error	*p* Value (IC 95%)
Li	0.013–0.345	0.08 ± 0.07	0.013	<0.010 (0.052–0.110)
Mg	146.640–3229.420	900.0 ± 700.0	132.348	<0.010 (660.291–1200.873)
Al	0–509.800	100.00 ± 100.00	22.481	<0.010 (48.890–140.740)
Ni	0.095–2.806	0.5 ± 0.5	0.095	<0.010 (0.339–0.727)
Cu	1.880–216.030	40.0 ± 50.0	9.54	<0.010 (16.11–55.08)
As	0–1.769	0.3 ± 0.4	0.074	<0.010 (0.096–0.402)
Ag	0–0.752	0.1 ± 0.2	0.031	<0.01 (0.028–0.156)
Cd	0–4.534	1 ± 1	0.184	<0.010 (0.210–0.963)
W	0.112–1.430	0.4 ± 0.3	0.054	<0.010 (0.2769–0.495)
TI	0.005–0.061	0.01 ± 0.01	0.002	<0.010 (0.009–0.018)
Pb	0.011–0.754	0.1 ± 0.2	0.035	<0.010 (0.081–0.223)
Bi	0.010–0.214	0.05 ± 0.05	0.008	<0.010 (0.039–0.070)
U	0.001–0.052	0.01 ± 0.01	0.003	<0.010 (0.007–0.017)
Be	0.010–0.050	0.01 ± 0.01	0.001	<0.010 (0.010–0.019)
Sb	0.010–0.057	0.02 ± 0.01	0.002	<0.010 (0.015–0.023)
Pt	0.005–0.032	0.01 ± 0.01	0.001	<0.010 (0.007–0.011)
Hg	0.017–0.659	0.1 ± 0.2	0.028	<0.01 (0.067–0.183)

**Table 4 toxics-12-00741-t004:** Trace element quantities found in products according to their origin.

Element (ppm)	America *n* = 5		Asia *n* = 11		Europe *n* = 15		*p* Value
	Mean ± Standard Error	*p* Value (IC 95%)	Media ± Standard Error	*p* Value (IC 95%)	Media ± Standard Error	*p* Value (IC 95%)	
Li	0.11 ± 0.02	0.011 (0.041–0.173) b	0.11 ± 0.03	0.005 (0.041–0.176) b	0.039 ± 0.007	<0.001 (0.023–0.055) a	0.026
Mg	516.4 ± 60.0	<0.001 (353.530–679.190) a	1052.68 ± 300.00	0.006 (386.810–1718.561) b	979.11 ± 200.00	<0.001 (637.090–1321.120) b	0.391
Al	100.6 ± 10.0	0.002 (64.480–136.680) b	144.3 ± 10.0	0.006 (52.930–235.120) c	56.9 ± 30.0	0.111 (–14.790–128.520) a	0.218
Ni	0.5 ± 0.1	0.012 (0.175–0.791) a	0.5 ± 0.1	0.007 (0.172–0.818) a	0.6 ± 0.2	0.004 (0.221–0.935) b	0.907
Cu	14.4 ± 0.6	<0.001 (12.666–16.231) a	79.7 ± 20.0	0.004 (31.960–127.411) b	10 ± 2	<0.001 (5.740–14.890) a	<0.001
As	0.08 ± 0.02	0.009 (0.032–0.122) a	0.5 ± 0.2	0.014 (0.134–0.948) b	0.11 ± 0.02	<0.001 (0.073–0.146) a	0.012
Ag	0.016 ± 0.004	0.011 (0.006–0.027) a	0.24 ± 0.07	0.007 (0.081–0.395) b	0.013 ± 0.001	<0.001 (0.011–0.016) a	<0.001
Cd	0.12 ±0.03	0.012 (0.043–0.192) a	1.5 ± 0.4	0.004 (0.577–2.374) b	0.10 ± 0.04	0.018 (0.019–0.176) a	<0.001
W	0.39 ± 0.06	0.002 (0.234–0.547) b	0.5 ± 0.1	0.003 (0.212–0.803) c	0.29 ± 0.04	<0.001 (0.204–0.386) a	0.204
TI	0.015 ± 0.003	0.005 (0.007–0.023) b	0.016 ± 0.005	0.009 (0.005–0.027) b	0.009 ± 0.001	<0.001 (0.007–0.012) a	0.287
Pb	0.13 ± 0.02	0.002 (0.081–0.183) b	0.21 ± 0.08	0.005 (0.111–0.470) c	0.06 ± 0.02	0.002 (0.024–0.091) a	0.006
Bi	0.050 ± 0.009	0.004 (0.026–0.074) b	0.07 ± 0.02	0.005 (0.027–0.120) c	0.039 ± 0.007	<0.001 (0.024–0.053) a	0.170
U	0.0040 ± 0.0001	<0.001 (0.003–0.005) a	0.021 ± 0.006	0.003 (0.008–0.034) b	0.007 ± 0.002	<0.001 (0.003–0.011) a	0.011
Be	0.011 ± 0.001	<0.001 (0.009–0.014) a	0.011 ± 0.001	<0.001 (0.090–0.014) a	0.013 ± 0.003	< 0.001 (0.007–0.019) b	0.839
Sb	0.022 ± 0.001	<0.001 (0.019–0.025) b	0.017 ± 0.002	<0.001 (0.013–0.022) a	0.016 ± 0.003	<0.001 (0.01–0.023) a	0.529
Pt	0.009 ± 0.001	0.00 (0.006–0.12) a	0.009 ± 0.001	<0.001 (0.006–0.012) a	0.009 ± 0.002	<0.001 (0.005–0.013) a	0.979
Hg	0.06 ± 0.01	0.001 (0.033–0.089) a	0.25 ± 0.06	0.003 (0.109–0.394) b	0.05 ± 0.01	<0.001 (0.031–0.076) a	0.002

Different letters in rows indicate statistically significant differences among samples for the same element.

## Data Availability

The original contributions presented in the study are included in the article, further inquiries can be directed to the corresponding authors.

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
