# Peer review of "Presence of Trace Elements in Edible Insects Commercialized through Online E-Commerce Platform"

_toxics, 2024, doi:10.3390/toxics12100741_

Round 1
Reviewer 1 Report
Comments and Suggestions for Authors
The manuscript (toxics-3209536) presents the studies concerning the presence of trace elements in edible insects commercialized through online e-commerce platform. The topic raised by the authors is important and necessary. However, awareness of the consumption of insects as a substitute for commonly consumed meat is still negligible, as observed by the authors. The work has a rather thoughtful, balanced and clear structure. The discussion of the results is well developed and supported by the authors' own reflections based on other publications. Nevertheless, I have the following objections to the work:
1) Was it only from the Amazon platform that the insects were purchased? Because not all products displayed. At what time was the product purchase made? Please add time of purchase.
2) Is there data/numbers available in what quantities of edible insects are sold on the platform mentioned in the title?
3) Please complete the introduction with the importance of transition metals in food. Which are unacceptable, etc.
4) What atm pressure was used during insect mineralization?
5) In the Results section in the name mapane worms should be mopane worms.
6) In the Results section, the authors wrote that the CJ20 sample showed high levels of nitrogen and coal..... The work was supposed to be about trace elements. Also, why coal and not carbon? Please correct this.
7) I propose to standardize the unit and give the results as ppm.
8) There is no drawing or graph showing the elemental content of the insects studied.
9) Minor errors:
Line 241: …which It is gaining…”
line 303 => remove full stop in middle of sentence.
Line 349: “…MG, Cd and Ag. For H. If…”?????
Author Response
Reviewer’s comment: The manuscript (toxics-3209536) presents the studies concerning the presence of trace elements in edible insects commercialized through online e-commerce platform. The topic raised by the authors is important and necessary. However, awareness of the consumption of insects as a substitute for commonly consumed meat is still negligible, as observed by the authors. The work has a rather thoughtful, balanced and clear structure. The discussion of the results is well developed and supported by the authors' own reflections based on other publications. Nevertheless, I have the following objections to the work: Was it only from the Amazon platform that the insects were purchased? Because not all products displayed. At what time was the product purchase made? Please add time of purchase.
Author’s comment: All the insects were purchased from Amazon, and the purchase date was in December 2021. This information has been added to the text, but we avoided using the specific brand name "Amazon" to prevent any potential legal or trademark issues, especially if there is no official affiliation or endorsement from the company. Instead, it is advisable to refer to it as "an online e-commerce platform" to keep the focus on the general concept of online marketplaces rather than a specific brand. This approach helps avoid any implications of endorsement, association, or use of a trademarked name without permission, which could raise concerns in a formal or academic context.
Reviewer’s comment: Is there data/numbers available in what quantities of edible insects are sold on the platform mentioned in the title?
Author’s comment: There is none. Normally, that information is held by the distributor through their platform, but they did not provide us with that data.
Reviewer’s comment: Please complete the introduction with the importance of transition metals in food. Which are unacceptable, etc.
Author’s comment: Transition metals in food play a crucial role in human nutrition, as some are essential for various biological processes, while others can be toxic or harmful. Essential transition metals, such as iron (Fe), zinc (Zn), and copper (Cu), are vital micronutrients. They are involved in numerous physiological functions; iron is necessary for oxygen transport and energy metabolism; zinc supports the immune system and aids in cell division and growth and copper is involved in red blood cell formation and maintaining nerve cells and the immune system. However, excessive levels of these metals can cause toxicity. For instance, excess iron can lead to organ damage, and high copper intake may result in liver and kidney damage. On the other hand, some transition metals are considered toxic and unacceptable in food, even at low concentrations. These include, among others, cadmium; which can cause kidney damage and skeletal issues, lead; known for its neurotoxic effects, particularly in children, and mercury; which affects the nervous system and is particularly harmful during pregnancy.
It has been added in the text.
Reviewer’s comment: What atm pressure was used during insect mineralization?
Author’s comment: The value of atm pressure measured was 1 atm
Reviewer’s comment: In the Results section in the name mapane worms should be mopane worms.
Author’s comment: According to your comment, It has been modified in the text.
Reviewer’s comment: In the Results section, the authors wrote that the CJ20 sample showed high levels of nitrogen and coal..... The work was supposed to be about trace elements. Also, why coal and not carbon? Please correct this.
Author’s comment: It has been corrected.
Reviewer’s comment: I propose to standardize the unit and give the results as ppm.
Author’s comment: According to your comment, it has standardized.
Reviewer’s comment: There is no drawing or graph showing the elemental content of the insects studied.
Author’s comment: Previously to write this manuscript, authors reviewed this possibility but the idea was discarded because it wasn't visual. Instead, it was preferred to present the values in tables.
Reviewer’s comment: Minor errors: Line 241: …which It is gaining…”
Author’s comment: It has been corrected.
Reviewer’s comment: Line 303 => remove full stop in middle of sentence.
Author’s comment: It has been removed.
Reviewer’s comment: Line 349: “…MG, Cd and Ag. For H. If…”?????
Author’s comment: It has been corrected.
Reviewer 2 Report
Comments and Suggestions for Authors
The manuscript refers to the Presence of Trace Elements in Edible Insects Commercialized through Online E-Commerce Platform. The ms is interesting, but in the present form it cannot be accepted. I suggest following the first general advice and resubmit the ms.
General advice
In Table 2, there is only one number for each element, without a standard deviation. This means that author performed only ONE analysis for each sample? This is not scientifically acceptable. To be able to compare the data statistically, at least 3 measurements are needed to have a standard deviation. In addition, three different aliquots should be analysed for each sample and three analyses applied to each of these, so that a total of nine measurements can be obtained, in order to carry out a robust statistical analysis. So, in order to continue the ms revision you need to have at least 3 measurements for each sample.
Not all samples contain 100% of insects, so the content of elements also depends on other ingredients. Has this been taken into account? The authors do not comment on this.
The term “heavy metals” is outdated. Moreover, some elements are not metals (e.g As). I suggest using “trace elements” or “potentially toxic elements”.
Specific advice
Line 27. Current
Line 30. I suggest including in the references the EU regulation where insects are indicated as Novel Food (EU 2015/2283).
Line 39-41. List the scientific names (in italics) of all insects mentioned (migratory locust = Locusta migratoria).
Line 70-71. Edible insects and insect-based food. Please specify the number of acquired samples and refer here to Table 1.
Line 72: “to reduce moisture content” is not a scientific definition: I think it is necessary to completely eliminate water from the sample, because you give results in ug/kg dry weight.
Line 77-78. Did authors perform the analysis only in one aliquot for each sample? From the statistical point of view, it is better to perform analysis on three aliquots per sample.
Line 83: 0.20-0.40 g for each sample is a quantity different form 50 mg (line 77). Please explain.
Table 2: the first column with sample name is lacking. Moreover, there are only 20 rows, whereas the number of samples is 31. Please correct
Line 227: Poma et al demonstrated that….

Moderate editing of English language required.
Author Response
Reviewer’s comment: The manuscript refers to the Presence of Trace Elements in Edible Insects Commercialized through Online E-Commerce Platform. The ms is interesting, but in the present form it cannot be accepted. I suggest following the first general advice and resubmit the ms.
General advice
In Table 2, there is only one number for each element, without a standard deviation. This means that author performed only ONE analysis for each sample? This is not scientifically acceptable. To be able to compare the data statistically, at least 3 measurements are needed to have a standard deviation. In addition, three different aliquots should be analysed for each sample and three analyses applied to each of these, so that a total of nine measurements can be obtained, in order to carry out a robust statistical analysis. So, in order to continue the ms revision you need to have at least 3 measurements for each sample.
Author’s comment: Sorry for this mistake. We thought that it is necessary to include the mean, but it is confusing. We have re-written this paragraph to indicate three measurements for each sample. We have taken longer to respond to this issue because we needed to review the standard deviation data with the team and include it in the table.
Reviewer’s comment: Not all samples contain 100% of insects, so the content of elements also depends on other ingredients. Has this been taken into account? The authors do not comment on this.
Author’s comment: Thank you for your observation. You are correct in noting that not all samples contain 100% insects. However, our study focuses on analyzing the products that include insects, regardless of their percentage of insect content. While the presence of other ingredients could influence the element concentrations, the aim of our research is to evaluate the overall product that contains insects as part of its composition. This approach reflects how these products are commercially available to consumers.
Reviewer’s comment: The term “heavy metals” is outdated. Moreover, some elements are not metals (e.g As). I suggest using “trace elements” or “potentially toxic elements”.
Author’s comment: According to your comment, we have replaced with “trace elements”.
Reviewer’s comment: Specific advice: Line 27. Current
Author’s comment: It has been modified.
Reviewer’s comment: Line 30. I suggest including in the references the EU regulation where insects are indicated as Novel Food (EU 2015/2283).
Author’s comment: It has been added.
Reviewer’s comment: Line 39-41. List the scientific names (in italics) of all insects mentioned (migratory locust = Locusta migratoria).
Author’s comment: According to your comment, all insects are mentioned as scientific names.
Reviewer’s comment: Line 70-71. Edible insects and insect-based food. Please specify the number of acquired samples and refer here to Table 1.
Author’s comment: According to your comment, it has been added.
Reviewer’s comment: Line 72: “to reduce moisture content” is not a scientific definition: I think it is necessary to completely eliminate water from the sample, because you give results in ug/kg dry weight.
Author’s comment: According to your comment, it has been modified.
Reviewer’s comment: Line 77-78. Did authors perform the analysis only in one aliquot for each sample? From the statistical point of view, it is better to perform analysis on three aliquots per sample.
Author’s comment: As previously mentioned, we had not expressed it correctly. It has been modified in the text for clarity.
Reviewer’s comment: Line 83: 0.20-0.40 g for each sample is a quantity different form 50 mg (line 77). Please explain.
Author’s comment: As previously mentioned, we had not expressed it correctly. It has been modified in the text for clarity.
Reviewer’s comment: Table 2: the first column with sample name is lacking. Moreover, there are only 20 rows, whereas the number of samples is 31. Please correct
Author’s comment: According to your comment, it has been modified.
Reviewer’s comment: Line 227: Poma et al demonstrated that….
Author’s comment: According to your comment, it has been modified.
Reviewer 3 Report
Comments and Suggestions for Authors
This study (Presence of Trace Elements in Edible Insects Commercialized through Online E-Commerce Platform) investigates the presence of various elements in edible insect-based food products available for human consumption. The research involved analyzing several products using spectroscopy techniques and performing descriptive statistical analysis with IBM SPSS Statistics 27.
In my opinion, the topic addressed is very interesting. The authors precisely highlight how the work in question was carried out. Overall, the study is well-conducted; however, there remain some concerns that I will outline in the following comments, which are divided into general and specific remarks.
- From an analytical standpoint, the work should have been supported by an optimization of the method, for instance, using certified material. Has this part been carried out?
- In the abstract, you mention that 'Several products were analyzed using atomic and molecular spectroscopy.' However, only ICP-MS was used, so 'molecular' is not accurate.
- Pay attention to certain nomenclature: for example, sometimes 'digestion' is used, and other times 'mineralization'. In my opinion, the latter is preferable, but the important thing is consistency. The same applies to deionized water, where different terms like 'distilled', 'ultrapure', and 'deionised' are used. Please standardize.
- The title of the paragraph "2.3. Digestion and analytical preparation, and chemical analysis" doesn't seem correct to me, or I would consider revising it.
- In the results section, in my opinion, some points are not very clear. For example, at line 125 you mention, 'Some samples, particularly CJ20 (flour from Belgium), exhibit exceptionally high levels of nitrogen and coal, which may indicate contamination.' Firstly, the term 'coal' is incorrect. Who conducted these analyses? I do not see them in the results; or did you mean that these analyses were performed by others?
- Table 2 needs to be revised. The sample name definitely needs to be added. Furthermore, were the analyses performed in single measurements? I don't see a standard deviation, which could be a limitation of the study.
- Why does the sample weight have such a variable range from 0.2 to 0.4 g? Furthermore, on what basis was the digestion mixture selected? Did you perform a method optimization, or did you rely on other studies?
- It would be advisable to specify which isotopes were analyzed using ICP-MS.
- Finally, I would add a comparison with legal limits concerning potentially toxic element concentrations and expand the bibliography with some recent similar studies to make comparisons. Here are a few examples:
10.1093/jaoacint/qsad083
10.1007/s12011-022-03462-6
10.3390/molecules29163845
10.3920/JIFF2021.0068
Author Response
Reviewer’s comment: This study (Presence of Trace Elements in Edible Insects Commercialized through Online E-Commerce Platform) investigates the presence of various elements in edible insect-based food products available for human consumption. The research involved analyzing several products using spectroscopy techniques and performing descriptive statistical analysis with IBM SPSS Statistics 27.
In my opinion, the topic addressed is very interesting. The authors precisely highlight how the work in question was carried out. Overall, the study is well-conducted; however, there remain some concerns that I will outline in the following comments, which are divided into general and specific remarks.
- From an analytical standpoint, the work should have been supported by an optimization of the method, for instance, using certified material. Has this part been carried out?
Author’s comment: In the optimization of the method, no certified material has been used, as these are very specific matrices. The analytical laboratory used an internal method which is useful in the analysis of these compounds. Examples of internal method are reflected in the literature:
- Zhang, C., Li, J., Dai, Y., Gustave, W., Zhai, W., Zhong, Z., & Chen, J. (2023). Spatial and Temporal Variations of Heavy Metals’ Bioavailability in Soils Regulated by a Combined Material of Calcium Sulfate and Ferric Oxide. Toxics, 11(4), 296.
- Zhang, C., Li, J., Dai, Y., Gustave, W., Zhai, W., Zhong, Z., & Chen, J. (2023). Spatial and Temporal Variations of Heavy Metals’ Bioavailability in Soils Regulated by a Combined Material of Calcium Sulfate and Ferric Oxide. Toxics, 11(4), 296.
- Bebek Markovinović, A., Putnik, P., Duralija, B., Krivohlavek, A., Ivešić, M., Mandić Andačić, I., ... & Bursać Kovačević, D. (2022). Chemometric valorization of strawberry (Fragaria x ananassa duch.) cv.‘Albion’for the production of functional juice: The impact of physicochemical, toxicological, sensory, and bioactive value. Foods, 11(5), 640.
Reviewer’s comment: In the abstract, you mention that 'Several products were analyzed using atomic and molecular spectroscopy.' However, only ICP-MS was used, so 'molecular' is not accurate.
Author’s comment: According to your comment, it has been modified.
Reviewer’s comment: Pay attention to certain nomenclature: for example, sometimes 'digestion' is used, and other times 'mineralization'. In my opinion, the latter is preferable, but the important thing is consistency. The same applies to deionized water, where different terms like 'distilled', 'ultrapure', and 'deionised' are used. Please standardize.
Author’s comment: According to your comment, we have changed your idea in the manuscript.
Reviewer’s comment: The title of the paragraph "2.3. Digestion and analytical preparation, and chemical analysis" doesn't seem correct to me, or I would consider revising it.
Author’s comment: According to your comment, this title has been changed.
Reviewer’s comment: In the results section, in my opinion, some points are not very clear. For example, at line 125 you mention, 'Some samples, particularly CJ20 (flour from Belgium), exhibit exceptionally high levels of nitrogen and coal, which may indicate contamination.' Firstly, the term 'coal' is incorrect. Who conducted these analyses? I do not see them in the results; or did you mean that these analyses were performed by others?
Author’s comment: Thank you for your observation. You are correct, the term "coal" was incorrectly used. It should have been "carbon." This has now been corrected in the manuscript. Regarding the analyses, all the analyses mentioned in the study, including those for nitrogen and carbon, were conducted by our team as part of the research. We apologize for any confusion caused by the wording, and we have clarified this in the revised version of the text.
Reviewer’s comment: Table 2 needs to be revised. The sample name definitely needs to be added. Furthermore, were the analyses performed in single measurements? I don't see a standard deviation, which could be a limitation of the study.
Author’s comment: Table 2 has been reviewed. We apologize for the delay in responding to your comment, as we had to re-extract the data from the equipment and calculate the RSD value.
Reviewer’s comment: Why does the sample weight have such a variable range from 0.2 to 0.4 g? Furthermore, on what basis was the digestion mixture selected? Did you perform a method optimization, or did you rely on other studies?
Author’s comment: According to your comment, we have clarified this value in the manuscript. On the other hand, the method optimization was carried out with the Truzzi et al. (2019) (Truzzi, C.; Illuminati, S.; Girolametti, F.; Antonucci, M.; Scarponi, G.; Ruschioni, S.; Riolo, P.; Annibaldi, A. Influence of feeding substrates on the presence of toxic metals (Cd, Pb, Ni, As, Hg) in larvae of Tenebrio molitor: Risk assessment for human con-sumption. Int. J. Environ. Res. Public Health 2019, 16, 4815. https://doi.org/10.3390/ijerph16234815). This reference has been incorporated in the method procedure section.
Reviewer’s comment: It would be advisable to specify which isotopes were analyzed using ICP-MS.
Author’s comment: According to your comment, we have added isotopes used in this study.
Reviewer’s comment: Finally, I would add a comparison with legal limits concerning potentially toxic element concentrations and expand the bibliography with some recent similar studies to make comparisons. Here are a few examples:
10.1093/jaoacint/qsad083
10.1007/s12011-022-03462-6
10.3390/molecules29163845
10.3920/JIFF2021.0068
Author’s comment: According to your comment, these references have been added.
Round 2
Reviewer 1 Report
Comments and Suggestions for Authors
In my opinion, the manuscript in its current form is ready for publication.
Author Response
Reviewer’s comment: In my opinion, the manuscript in its current form is ready for publication.
Author’s comment: Thank you for your comment.
Reviewer 2 Report
Comments and Suggestions for Authors
The manuscript refers to the Presence of Trace Elements in Edible Insects Commercialized through Online E-Commerce Platform. The ms is interesting, but in the present form it cannot be accepted. I suggest major revision following some advice.
General advice
1) The principal problem of this ms is that results are reported in a bad way. It is difficult to follow the text that explains data in Tables for the following reasons:
- Some data present in the text are not reported in tables (see lines 123-144)
- There are differences among data reported in the text and in Tables: Lines 196-197: the cited regions (and the number of samples for each one) do not correspond to those present in Table 4. In the text authors wrote: Asia (n=8), Europe (n=8) United Kingdom (n=15); in Table 4: Asia (n=11), Europe (n=15) America (n=5).
- Data reported in the discussion section are not reported in Tables.
- Moreover, the discussion is focused on discussing the presence of trace elements in samples divided by insect species, but the results are not presented in such way, so it is very difficult to follow the discussion. I suggest presenting in Table 2 samples following the insect species, so first all samples containing the insect a, then all samples containing the insect b, and so on. Moreover, at the end of samples containing a specific insect species, authors should report a raw with the mean content and SD for each element.
- The statistical analysis is not well done. Authors should add letters after number in tables, indicating statistically significant differences. In the Results section, instead of reporting data already present in tables, authors should talk about statistically significant differences among samples containing the same insect species, or coming form different countries.
2) Pay attention to the significant figures. The general rule is that experimental uncertainty (Standard Deviation) should be rounded up to one significant figure (which would be the uncertain figure) and the mean must be rounded at the same figure. In the text authors reported too many digits. For example, replace, “1337.61±0.3” with “1337.6±0.3”, or “1902.74±2.3” with “1903±2”, and so on. In Table 3 e.g., replace “0.081±0.074” with “0.08±0.07”, or 0.249±0.417 with 0.2±0.4). Please correct anywhere in all the tables. Moreover, in Table 3 and in Table 4 the SD is sometimes higher than the mean. I suggest using the standard error instead of the standard deviation when there are a high number of samples with so different concentrations of trace elements.
3) In Table 2, standard deviation has been added, but it is very strange that in some cases le RSD is only 0.02% or 0.1% (see for example the case of Mg). Please check SD
4) About my previous suggestion “Not all samples contain 100% of insects, so the content of elements also depends on other ingredients. Has this been taken into account?”, authors answered that the study focuses on analyzing the products that include insects, regardless of their percentage of insect content. While the presence of other ingredients could influence the element concentrations, the aim of the research is to evaluate the overall product that contains insects as part of its composition. This is ok, but the problem is that in the discussion section authors compare their results on insect-based food products (that sometimes have a very low percentage of insect, such as CJ14 or CJ22, for example) with trace elements reported in literature for insects, and this is not correct.
Specific advice
Line 30. I suggest, as already done, including in the references the EU regulation where insects are indicated as Novel Food (EU 2015/2283). Authors did not add this reference in the text, where they indicated insects as Novel Food for the first time. This important reference is cited only in the conclusion section (where it is well known that no references should be added)
Line 83. You did not collect only edible insects, but also insect-based food. Please add in the text and specify the number of acquired samples (31) and refer here to Table 1. Why in Table 1 the samples are coded CJ…? I think it is better to call samples as 1, 2, 3, and so on. Or, better, the code should contain the initials of the insect’s name present in the sample, followed by a number (e.g. for Acheta domesticus, AD1, AD2, and so on..)
Line 91: authors wrote “Insect powder was used for each sample analysis and all analyses were performed in triplicate”. Do this mean that authors performed analyses in three aliquots per sample or that they performed 3 measurements for each sample with the instrumentation? I suppose that you performed analysis in one aliquot per sample, and that you performed three measurements for each sample. Please specify better.
Lines 123-144. I do not understand which results author refer. Authors said “The analysis of various edible insect products, as detailed in Table 1, reveals significant findings regarding the presence of essential and non-essential metals, as well as a mineral. But in Table 1 there are not these data”.
Line 138: high levels of Nitrogen and Carbon??? Where are these data? How did authors perform this analysis?
Table 2: in the first column I suggest inserting the code of the sample, instead of the name, because in the text authors describe the trace elements content naming sample with the code.
Line 163: .. different edible insect samples and insect-based food products
Lines 162-165: this sentence should be moved to the Discussion section
Lines 172-189: data are already present in Table 3, and it is boring to read this text. Authors should comment on the results, and not just reporting data already present in the table.
Lines 189-191: this sentence is a repetition of the phrase already present in lines 162-165 /that should be moved to the discussion section).
Lines 191-194: should be moved in the discussion section.
Line 186: edible insect and insect-based food products
Table 3 name of the first column: Element (ppm). All the elements are measured in ppm, so put the unit of measure at the head of the column and delete (ppm) in the rows. Moreover, in table authors write ppm, and in the text µg/kg. It is the same unit, but please use the same way to write it.
Lines 196-197: the cited regions (and the number of samples for each one) do not correspond to those present in Table 4. In the text authors wrote: Asia (n=8), Europe (n=8) United Kingdom (n=15); in Table 4: Asia (n=11), Europe (n=15) America (n=5).
Table 4: replace “P Valor” with “P value”
Lines 201-223: data reported in the text do not correspond to data reported in Table 4.
Lines 224-227: the same sentence already reported in lines 162-165 and in lines 189-191.
Lines 282-306: are these data referred to Acheta domesticus?
Line 322: delete the repetition “trace elements”
Lines 394-397: the sentence needs a reference.

Extensive editing of English language required.
Author Response
Reviewer’s comment: The manuscript refers to the Presence of Trace Elements in Edible Insects Commercialized through Online E-Commerce Platform. The ms is interesting, but in the present form it cannot be accepted. I suggest major revision following some advice.
General advice
- The principal problem of this ms is that results are reported in a bad way. It is difficult to follow the text that explains data in Tables for the following reasons:
Some data present in the text are not reported in tables (see lines 123-144)
There are differences among data reported in the text and in Tables: Lines 196-197: the cited regions (and the number of samples for each one) do not correspond to those present in Table 4. In the text authors wrote: Asia (n=8), Europe (n=8) United Kingdom (n=15); in Table 4: Asia (n=11), Europe (n=15) America (n=5).
Author’s comment: With your suggestions and those of the other reviewers that we have now implemented, we believe the manuscript is better adapted. We appreciate your help with each comment, which we have adjusted according to your recommendations.
Reviewer’s comment: - Data reported in the discussion section are not reported in Tables.
Author’s comment: We have followed the suggestions of the other two reviewers, and the format of the discussion section makes sense in this context to explain the species of each sample, providing more clarity and reliability to the data.
Reviewer’s comment: - Moreover, the discussion is focused on discussing the presence of trace elements in samples divided by insect species, but the results are not presented in such way, so it is very difficult to follow the discussion. I suggest presenting in Table 2 samples following the insect species, so first all samples containing the insect a, then all samples containing the insect b, and so on. Moreover, at the end of samples containing a specific insect species, authors should report a raw with the mean content and SD for each element.
Author’s comment: We appreciate your suggestion regarding organizing the samples by insect species in Table 2. We understand that this approach could make the discussion easier to follow. However, our current layout is designed to provide an overview based on the type of product rather than the insect species, as we believe this better reflects the market analysis of the products being evaluated. This arrangement allows for a comparison of products as they are commercially presented, which is a key objective of our study. Nonetheless, we value your input and will consider including a more detailed species-specific analysis in future studies. Thank you for your understanding regarding this approach. On the other hand, the other two reviewers indicate that the way it is written is fine.
Reviewer’s comment: - The statistical analysis is not well done. Authors should add letters after number in tables, indicating statistically significant differences. In the Results section, instead of reporting data already present in tables, authors should talk about statistically significant differences among samples containing the same insect species, or coming form different countries.
Author’s comment: We appreciated your suggestions and implemented the following changes to improve the presentation and interpretation of our results. We added letters after the numbers in our tables to indicate statistically significant differences between samples. This made it easier for readers to quickly identify significant variations in our data. We have taken your suggestion into account not to repeat the data already present in the tables, and we have modified the corresponding text.
Reviewer’s comment: 2) Pay attention to the significant figures. The general rule is that experimental uncertainty (Standard Deviation) should be rounded up to one significant figure (which would be the uncertain figure) and the mean must be rounded at the same figure. In the text authors reported too many digits. For example, replace, “1337.61±0.3” with “1337.6±0.3”, or “1902.74±2.3” with “1903±2”, and so on. In Table 3 e.g., replace “0.081±0.074” with “0.08±0.07”, or 0.249±0.417 with 0.2±0.4). Please correct anywhere in all the tables. Moreover, in Table 3 and in Table 4 the SD is sometimes higher than the mean. I suggest using the standard error instead of the standard deviation when there are a high number of samples with so different concentrations of trace elements.
Author’s comment: We have unified the format of all numbers in the manuscript, presenting them with 3 decimal places to enhance the precision of our research findings. We have added the standard error values in Tables 3 and 4.
Reviewer’s comment: In Table 2, standard deviation has been added, but it is very strange that in some cases le RSD is only 0.02% or 0.1% (see for example the case of Mg). Please check SD
Author’s comment: We have checked these values and they are OK in the Table 2.
Reviewer’s comment: About my previous suggestion “Not all samples contain 100% of insects, so the content of elements also depends on other ingredients. Has this been taken into account?”, authors answered that the study focuses on analyzing the products that include insects, regardless of their percentage of insect content. While the presence of other ingredients could influence the element concentrations, the aim of the research is to evaluate the overall product that contains insects as part of its composition. This is ok, but the problem is that in the discussion section authors compare their results on insect-based food products (that sometimes have a very low percentage of insect, such as CJ14 or CJ22, for example) with trace elements reported in literature for insects, and this is not correct.
Author’s comment: We acknowledge the referee’s point regarding the comparison between insect-based products and pure insect data from the literature. However, our study's focus is to evaluate commercially available products, which contain varying percentages of insect content, as this reflects real-world consumer choices. While other ingredients may influence trace element concentrations, our objective is to analyze the overall composition of these products as they are presented in the market. This approach provides a more accurate representation of the products' nutritional and safety profiles.
Reviewer’s comment: Line 30. I suggest, as already done, including in the references the EU regulation where insects are indicated as Novel Food (EU 2015/2283). Authors did not add this reference in the text, where they indicated insects as Novel Food for the first time. This important reference is cited only in the conclusion section (where it is well known that no references should be added)
Author’s comment: They have been added. Now, they are references 6-9
Reviewer’s comment: Line 83. You did not collect only edible insects, but also insect-based food. Please add in the text and specify the number of acquired samples (31) and refer here to Table 1. Why in Table 1 the samples are coded CJ…? I think it is better to call samples as 1, 2, 3, and so on. Or, better, the code should contain the initials of the insect’s name present in the sample, followed by a number (e.g. for Acheta domesticus, AD1, AD2, and so on..)
Author’s comment: The type of sample has been added in the Table 1. CJ is an internal number in the laboratory. We have prefer maintained in this type due to your comment from later.
Reviewer’s comment: Line 91: authors wrote “Insect powder was used for each sample analysis and all analyses were performed in triplicate”. Do this mean that authors performed analyses in three aliquots per sample or that they performed 3 measurements for each sample with the instrumentation? I suppose that you performed analysis in one aliquot per sample, and that you performed three measurements for each sample. Please specify better.
Author’s comment: According to your comment, it has been carried out.
Reviewer’s comment: Lines 123-144. I do not understand which results author refer. Authors said “The analysis of various edible insect products, as detailed in Table 1, reveals significant findings regarding the presence of essential and non-essential metals, as well as a mineral. But in Table 1 there are not these data”.
Author’s comment: Sorry, it is a mistake. We have modified this paragraph.
Reviewer’s comment: Line 138: high levels of Nitrogen and Carbon??? Where are these data? How did authors perform this analysis?
Author’s comment: It has been deleted due to that do not be the aim of this study.
Reviewer’s comment: Table 2: in the first column I suggest inserting the code of the sample, instead of the name, because in the text authors describe the trace elements content naming sample with the code.
Author’s comment: According to your comment, it has been carried out.
Reviewer’s comment: Line 163: .. different edible insect samples and insect-based food products
Author’s comment: According to your comment, it has been carried out.
Reviewer’s comment: Lines 162-165: this sentence should be moved to the Discussion section
Author’s comment: According to your comment, it has been carried out.
Reviewer’s comment: Lines 172-189: data are already present in Table 3, and it is boring to read this text. Authors should comment on the results, and not just reporting data already present in the table.
Author’s comment: According to your comment, it has been carried out.
Reviewer’s comment: Lines 189-191: this sentence is a repetition of the phrase already present in lines 162-165 /that should be moved to the discussion section).
Author’s comment: According to your comment, it has been carried out.
Reviewer’s comment: Lines 191-194: should be moved in the discussion section.
Author’s comment: According to your comment, it has been transferred to discussion section.
Reviewer’s comment: Line 186: edible insect and insect-based food products
Author’s comment: According to your comment, it has been modified.
Reviewer’s comment: Table 3 name of the first column: Element (ppm). All the elements are measured in ppm, so put the unit of measure at the head of the column and delete (ppm) in the rows. Moreover, in table authors write ppm, and in the text µg/kg. It is the same unit, but please use the same way to write it.
Author’s comment: According to your comment, it has been replaced.
Reviewer’s comment: Lines 196-197: the cited regions (and the number of samples for each one) do not correspond to those present in Table 4. In the text authors wrote: Asia (n=8), Europe (n=8) United Kingdom (n=15); in Table 4: Asia (n=11), Europe (n=15) America (n=5).
Author’s comment: According to your comment, it has been replaced.
Reviewer’s comment: Table 4: replace “P Valor” with “P value”
Author’s comment: According to your comment, it has been replaced.
Reviewer’s comment: Lines 201-223: data reported in the text do not correspond to data reported in Table 4.
Author’s comment: According to your comment, it has been modified.
Reviewer’s comment: Lines 224-227: the same sentence already reported in lines 162-165 and in lines 189-191.
Author’s comment: According to your comment, lines 162-165 and lines 189-191 have been deleted.
Reviewer’s comment: Lines 282-306: are these data referred to Acheta domesticus?
Author’s comment: Yes. Data are extracted from this reference… Ververis, E.; Boué, G.; Poulsen, M.; Pires, S. M.; Niforou, A.; Thomsen, S. T.; Tesson, V.; Federighi, M.; Naska, A. A systematic review of the nutrient composition, microbiological and toxicological profile of Acheta domesticus (house cricket). J. Food Compos. Anal. 2022, 114, 104859. https://doi.org/10.1016/j.jfca.2022.104859
Reviewer’s comment: Line 322: delete the repetition “trace elements”
Author’s comment: According to your comment, “trace elements” has been deleted.
Reviewer’s comment: Lines 394-397: the sentence needs a reference.
Author’s comment: According to your comment, it has been added.
Reviewer’s comment: Extensive editing of English language required.
Author’s comment: According to your comment, a native-English professional has been reviewed the manuscript.
Reviewer 3 Report
Comments and Suggestions for Authors
The authors have satisfied their requests to a fairly complete extent.
Author Response
Reviewer’s comment: The authors have satisfied their requests to a fairly complete extent.
Author’s comment: Thank you for your comment.
Round 3
Reviewer 2 Report
Comments and Suggestions for Authors
The manuscript refers to the Presence of Trace Elements in Edible Insects Commercialized through Online E-Commerce Platform. The ms is interesting, and it has been improved, thanks to referee suggestions. But further improvement is needed, following some advice.
1) Previous Reviewer’s comment: - Data reported in the discussion section are not reported in Tables. Author’s comment: We have followed the suggestions of the other two reviewers, and the format of the discussion section makes sense in this context to explain the species of each sample, providing more clarity and reliability to the data. Reviewer’s answer: I refer to data reported in the text that are different from data reported in Tables. E.g. see lines 377-379: “an average lead concentration of 0.129 ppm with a standard deviation of 0.044 ppm was recorded, while the average cadmium concentration was 0.128 ppm with a standard deviation of 0.03 ppm” in table 2 different mean values are reported for these two elements.
2) Previous Reviewer’s comment: - The statistical analysis is not well done. Authors should add letters after number in tables, indicating statistically significant differences. In the Results section, instead of reporting data already present in tables, authors should talk about statistically significant differences among samples containing the same insect species, or coming form different countries. Author’s comment: We appreciated your suggestions and implemented the following changes to improve the presentation and interpretation of our results. We added letters after the numbers in our tables to indicate statistically significant differences between samples. This made it easier for readers to quickly identify significant variations in our data. We have taken your suggestion into account not to repeat the data already present in the tables, and we have modified the corresponding text. Reviewer’s answer: authors should not put the letter “a” after the p-value. Please delete letter “a” after p-values because it has no sense. I suggested to put letters after numbers: in Table 2, different letters in columns indicate statistically significant differences among samples for the same element (i.e. I comparethe content of each element among different samples). In Table 4, different letters in rows indicate statistically significant differences among samples for the same element (i.e., I compare for each element its content among samples coming from different countries).
3) Previous Reviewer’s comment: Pay attention to the significant figures. The general rule is that experimental uncertainty (Standard Deviation) should be rounded up to one significant figure (which would be the uncertain figure) and the mean must be rounded at the same figure. In the text authors reported too many digits. For example, replace, “1337.61±0.3” with “1337.6±0.3”, or “1902.74±2.3” with “1903±2”, and so on. In Table 3 e.g., replace “0.081±0.074” with “0.08±0.07”, or 0.249±0.417 with 0.2±0.4). Please correct anywhere in all the tables. Author’s comment: We have unified the format of all numbers in the manuscript, presenting them with 3 decimal places to enhance the precision of our research findings. Reviewer’s answer: from the statistical point of view, it has no sense presenting all numbers with the same decimal place: read please the rules that I reported in my comment (see above).
4) Previous Reviewer’s comment: About my previous suggestion “Not all samples contain 100% of insects, so the content of elements also depends on other ingredients. Has this been taken into account?”, authors answered that the study focuses on analyzing the products that include insects, regardless of their percentage of insect content. While the presence of other ingredients could influence the element concentrations, the aim of the research is to evaluate the overall product that contains insects as part of its composition. This is ok, but the problem is that in the discussion section authors compare their results on insect-based food products (that sometimes have a very low percentage of insect, such as CJ14 or CJ22, for example) with trace elements reported in literature for insects, and this is not correct. Author’s comment: We acknowledge the referee’s point regarding the comparison between insect-based products and pure insect data from the literature. However, our study's focus is to evaluate commercially available products, which contain varying percentages of insect content, as this reflects real-world consumer choices. While other ingredients may influence trace element concentrations, our objective is to analyze the overall composition of these products as they are presented in the market. This approach provides a more accurate representation of the products' nutritional and safety profiles. Reviewer’s answer: if the focus is to evaluate commercially available products, which contain varying percentages of insect content, then in the discussion section authors should compare their results with element content of other insect-based food products, and not only with element content of insects.
Line 128: “replace “All edible insects” with “All edible insects and insect-based food (n=31)”
Line 129: After “December 2021” pleas add “(Table 1).
Line 136-137: replace “Insect powder was used for each sample analysis and performed on one aliquot per sample, and three measurements were taken for each sample” with “Insect powder was used for each sample analysis (one aliquot per sample)”. Move the sentence “three measurements were taken for each sample” to line 159.
Line 174-190: authors refer to “the findings”, but which findings do they refer? Results are reported in Table 2, and results are presented from line 212 to line 225. I suggest merging the two paragraph, reporting results in a better way.
Lines 220-224: CJ24 is not “tarantula” but “salt flavour Zebra Tarantula”. Please report the whole name for each sample, because authors said that they did not compare insects, but insect-based food. The same for CJ25: armor tail scorpions, not “scorpions”
Table 3: head of the firs column: Element (ppm), not only “ppm”
Comments on the Quality of English LanguageMinor editing of English language required.
Author Response
Reviewer’s comment: The manuscript refers to the Presence of Trace Elements in Edible Insects Commercialized through Online E-Commerce Platform. The ms is interesting, and it has been improved, thanks to referee suggestions. But further improvement is needed, following some advice.
1) Previous Reviewer’s comment: - Data reported in the discussion section are not reported in Tables. Author’s comment: We have followed the suggestions of the other two reviewers, and the format of the discussion section makes sense in this context to explain the species of each sample, providing more clarity and reliability to the data. Reviewer’s answer: I refer to data reported in the text that are different from data reported in Tables. E.g. see lines 377-379: “an average lead concentration of 0.129 ppm with a standard deviation of 0.044 ppm was recorded, while the average cadmium concentration was 0.128 ppm with a standard deviation of 0.03 ppm” in table 2 different mean values are reported for these two elements.
Author’s comment: We have deleted this part to avoid confusion.
Reviewer’s comment: 2) Previous Reviewer’s comment: - The statistical analysis is not well done. Authors should add letters after number in tables, indicating statistically significant differences. In the Results section, instead of reporting data already present in tables, authors should talk about statistically significant differences among samples containing the same insect species, or coming form different countries. Author’s comment: We appreciated your suggestions and implemented the following changes to improve the presentation and interpretation of our results. We added letters after the numbers in our tables to indicate statistically significant differences between samples. This made it easier for readers to quickly identify significant variations in our data. We have taken your suggestion into account not to repeat the data already present in the tables, and we have modified the corresponding text. Reviewer’s answer: authors should not put the letter “a” after the p-value. Please delete letter “a” after p-values because it has no sense. I suggested to put letters after numbers: in Table 2, different letters in columns indicate statistically significant differences among samples for the same element (i.e. I compare the content of each element among different samples). In Table 4, different letters in rows indicate statistically significant differences among samples for the same element (i.e., I compare for each element its content among samples coming from different countries).
Author’s comment: According to your comment, “a” after p-values have been deleted. In the other hand, in Table 2, we have added different letters in columns indicate statistically significant differences among samples for the same element (i.e. I compare the content of each element among different samples), being the format adapted in this Table 2 similar to Table 1 in this article: Mockevičiūtė, R.; Jurkonienė, S.; Šveikauskas, V.; Zareyan, M.; Jankovska-Bortkevič, E.; Jankauskienė, J.; Kozeko, L.; Gavelienė, V. Probiotics, Proline and Calcium Induced Protective Responses of Triticum aestivum under Drought Stress. Plants 2023, 12, 1301. https://doi.org/10.3390/plants12061301. Furthermore, in Table 4, we have added different letters in rows indicate statistically significant differences among samples for the same element (i.e., I compare for each element its content among samples coming from different countries), being the format adapted in this Table 4 similar to Table 1 in this article: Kowalczewski, P.Ł.; Radzikowska, D.; Ivanišová, E.; Szwengiel, A.; Kačániová, M.; Sawinska, Z. Influence of Abiotic Stress Factors on the Antioxidant Properties and Polyphenols Profile Composition of Green Barley (Hordeum vulgare L.). Int. J. Mol. Sci. 2020, 21, 397. https://doi.org/10.3390/ijms21020397
Reviewer’s comment: 3) Previous Reviewer’s comment: Pay attention to the significant figures. The general rule is that experimental uncertainty (Standard Deviation) should be rounded up to one significant figure (which would be the uncertain figure) and the mean must be rounded at the same figure. In the text authors reported too many digits. For example, replace, “1337.61±0.3” with “1337.6±0.3”, or “1902.74±2.3” with “1903±2”, and so on. In Table 3 e.g., replace “0.081±0.074” with “0.08±0.07”, or 0.249±0.417 with 0.2±0.4). Please correct anywhere in all the tables. Author’s comment: We have unified the format of all numbers in the manuscript, presenting them with 3 decimal places to enhance the precision of our research findings. Reviewer’s answer: Thank you for your comment. To unify the data according to the significant rules indicated by your comment, we need to follow these basic principles:
- The standard deviation (uncertainty) should be rounded to one significant figure.
- The mean value should be rounded to match the same decimal place as the standard deviation.
- Avoid presenting too many digits that don't contribute to useful precision, as it can be misleading from a statistical standpoint.
Examples:
Original value: 1337.61 ± 0.3
Correct unification: 1337.6 ± 0.3
Reason: The standard deviation has one significant figure (0.3), so the mean is adjusted to the same decimal place.
Original value: 1902.74 ± 2.3
Correct unification: 1903 ± 2
Reason: The standard deviation has two significant figures (2.3), so the mean is rounded to the same decimal place (whole number).
Original value: 0.081 ± 0.074
Correct unification: 0.08 ± 0.07
Reason: The standard deviation is rounded to one significant figure, and the mean is also adjusted to the same number of decimals.
Original value: 0.249 ± 0.417
Correct unification: 0.2 ± 0.4
Reason: The standard deviation is rounded to one significant figure (0.4), and the mean is also rounded to one decimal.
According to your idea and my comment, we have changed these values.
Reviewer’s comment: 4) Previous Reviewer’s comment: About my previous suggestion “Not all samples contain 100% of insects, so the content of elements also depends on other ingredients. Has this been taken into account?”, authors answered that the study focuses on analyzing the products that include insects, regardless of their percentage of insect content. While the presence of other ingredients could influence the element concentrations, the aim of the research is to evaluate the overall product that contains insects as part of its composition. This is ok, but the problem is that in the discussion section authors compare their results on insect-based food products (that sometimes have a very low percentage of insect, such as CJ14 or CJ22, for example) with trace elements reported in literature for insects, and this is not correct. Author’s comment: We acknowledge the referee’s point regarding the comparison between insect-based products and pure insect data from the literature. However, our study's focus is to evaluate commercially available products, which contain varying percentages of insect content, as this reflects real-world consumer choices. While other ingredients may influence trace element concentrations, our objective is to analyze the overall composition of these products as they are presented in the market. This approach provides a more accurate representation of the products' nutritional and safety profiles. Reviewer’s answer: if the focus is to evaluate commercially available products, which contain varying percentages of insect content, then in the discussion section authors should compare their results with element content of other insect-based food products, and not only with element content of insects.
Author’s comment: Thank you for your comment. We have added in the discussion section other literature about other insect-based food products.
Reviewer’s comment: Line 128: “replace “All edible insects” with “All edible insects and insect-based food (n=31)”
Author’s comment: According to your comment, it has been replaced.
Reviewer’s comment: Line 129: After “December 2021” pleas add “(Table 1).
Author’s comment: According to your comment, it has been added.
Reviewer’s comment: Line 136-137: replace “Insect powder was used for each sample analysis and performed on one aliquot per sample, and three measurements were taken for each sample” with “Insect powder was used for each sample analysis (one aliquot per sample)”. Move the sentence “three measurements were taken for each sample” to line 159.
Author’s comment: According to your comment, it has been carried out.
Reviewer’s comment: Line 174-190: authors refer to “the findings”, but which findings do they refer? Results are reported in Table 2, and results are presented from line 212 to line 225. I suggest merging the two paragraphs, reporting results in a better way.
Author’s comment: According to your comment, it has been carried out.
Reviewer’s comment: Lines 220-224: CJ24 is not “tarantula” but “salt flavour Zebra Tarantula”. Please report the whole name for each sample, because authors said that they did not compare insects, but insect-based food. The same for CJ25: armor tail scorpions, not “scorpions”
Author’s comment: According to your comment, it has been carried out.
Reviewer’s comment: Table 3: head of the firs column: Element (ppm), not only “ppm”
Author’s comment: According to your comment, it has been changed it.
Reviewer’s comment: Comments on the Quality of English Language. Minor editing of English language required.
Author’s comment: According to your comment, English-native person have been reviewed the manuscript.